# Flavonoid Nanoparticles: A Promising Approach for Cancer Therapy

**DOI:** 10.3390/biom10091268

**Published:** 2020-09-02

**Authors:** Malgorzata Dobrzynska, Marta Napierala, Ewa Florek

**Affiliations:** 1Department of Bromatology, Poznan University of Medical Sciences, 60-354 Poznan, Poland; mdobrzynska@ump.edu.pl; 2Laboratory of Environmental Research, Department of Toxicology, Poznan University of Medical Sciences, 60-631 Poznan, Poland

**Keywords:** flavonoid nanoparticles, flavonoid loaded nanobiomaterials, cancer, cancer treatments, anticancer activity

## Abstract

Flavonoids, a ubiquitous group of naturally occurring polyphenolic compounds, have recently gained importance as anticancer agents. Unfortunately, due to low solubility, absorption, and rapid metabolism of dietary flavonoids, their anticancer potential is not sufficient. Nanocarriers can improve the bioavailability of flavonoids. In this review we aimed to evaluate studies on the anticancer activity of flavonoid nanoparticles. A review of English language articles published until 30 June 2020 was conducted, using PubMed (including MEDLINE), CINAHL Plus, Cochrane, and Web of Science data. Most studies determining the anticancer properties of flavonoid nanoparticles are preclinical. The potential anticancer activity focuses mainly on MCF-7 breast cancer cells, A549 lung cancer cells, HepG2 liver cancer cells, and melanoma cells. The flavonoid nanoparticles can also support the anti-tumour effect of drugs used in cancer therapy by enhancing the anti-tumour effect or reducing the systemic toxicity of drugs.

## 1. Introduction

Flavonoids are the most common and widely distributed group of plant compounds, occurring virtually in all plant parts. This group can be divided into several subfamilies such as flavones, flavanols, flavanones, flavonols, and isoflavones [1]. As a dietary component, flavonoids are thought to have beneficial effects on human health [2]. Their health-promoting properties are associated with antioxidant [3], anti-inflammatory [4], and anticancer properties [5]. Nowadays attention is drawn to the anticancer activity of flavonoids. Many studies have documented flavonoids, for example, epigallocatechin-3-gallate, quercetin, genistein, apigenin, naringenin, silibinin, and kaempferol, to be effective against various types of cancer. Unfortunately, due to low solubility [2], poor absorption [6], and rapid metabolism [7], use of flavonoids in cancer treatment is not satisfactory. In this regard, modern nanotechnology may be used. Nanocarriers can improve the bioavailability of flavonoids [1]. In vitro and in vivo studies have shown potential anticancer activity of flavonoid nanoparticles against A549 lung cancer cells, B16F10 melanoma cells, MCF-7 breast cancer cells, HepG2 liver cancer cells or CT26 colorectal cancer cells [8,9,10,11,12]. There are several different types of flavonoid nanocarriers currently used in cancer therapy. These include polymeric nanoparticles [13], nanocapsules [14], metallic nanoparticles (gold) [15] or solid lipid nanocarriers [16].

This review aims to evaluate studies on the anticancer activity of flavonoid nanoparticles. A review of English language articles published until 30 June 2020 was conducted, using PubMed (including MEDLINE), CINAHL Plus (Cumulative Index to Nursing and Allied Health Literature), Cochrane, and Web of Science data. The literature search was performed using MESH terms (Medical Subject Headings) and other relevant keywords. The key search terms were “flavonoid nanoparticles”, “flavonoid loaded nanobiomaterials”, “flavonoid nanoparticles and cancer”; “flavonoid nanoparticles and cancer treatments”, and “flavonoid nanoparticles and anticancer activity”.

## 2. Epigallocatechin-3-gallate (EGCG)

Epigallocatechin-3-gallate is found in the leaves of green tea and its role in the treatment of cancer has been studied intensively [17,18]. The antioxidant effect of EGCG is limited by poor absorption after oral administration, unfavourable pharmacokinetics and biodistribution, low accumulation in the tissues, or low targeting efficacy [19]. It is worth noting that some studies suggest high doses of EGCG can induce toxicity in the liver [20]. However, EGCG nanoparticles induced high tumour growth inhibition in the case of prostate, breast, liver, gastric, bladder, and melanoma cancers. The concept of nanotherapy in cancer was introduced by Siddiqui et al. [21]. The polylactic acid-polyethylene glycol (PLA-PEG) with encapsulated EGCG nanoparticles were used in this study [21]. It was observed that these nanoparticles were more than 10-fold effective against 22Rr1 prostate cancer cells than free EGCG [21]. The use of PLA-PEG nanoparticles could be useful to limit the toxicity and enhance the bioavailability of EGCG. Injection EGCG PLA-PEG nanoparticles minimize carrier-induced undesirable cytotoxicity and improve the pharmacokinetic and pharmacodynamic properties of this flavonoid [21]. These positive results have initiated studies of other flavonoids and nanoparticles which may have potential effect in cancer treatment. Another example is EGCG-conjugated gold nanoparticles. The beneficial effect of gold nanoparticles in cancer treatment is associated with the small size, non-toxicity, and non-immunogenicity, which cause these nanoparticles to be able to pass natural barriers in human body and control the release of drugs in different locations. Furthermore, gold nanoparticles are mainly used for imaging and radiation sensitization [22]. In study by Hsieh et al., EGCG was physically attached to the surface of nanogold particles. In mice subcutaneously implanted with MBT-2 murine bladder tumour cells, an inhibition of growing cancer cells took place by means of cell apoptosis [23,24]. Following studies about EGCG gold-nanoparticles inserted into the tumour also confirmed a significant reduction of tumour growth in PC-3 prostate cells and B16F10 melanoma cells [25,26]. The Rocha study showed that the EGCG incorporated with a carbohydrate matrix of gum arabic and maltodextrin carbohydrate matrix induced apoptosis in Du145 prostate cancer cells [27]. The use of a carbohydrate matrix of arabic gum and maltodextrin nanoparticles contributes to preserve antioxidant properties and improves the bioavailability of flavonoids [28].

In the Siddiqui et al. study, EGCG-encapsulated chitosan induced apoptosis in Mel928 melanoma cells. Chitosan nanoparticles are characterized by mucoadhesive properties. After oral administration, flavonoids encapsulated in chitosan nanoparticles adhere to the gastrointestinal tract a for longer time, which cause a longer release of the drug [29]. Further research was carried out on prostate cancer cells. Oral administration of chitosan-based EGCG significantly inhibited prostate cancer cell growth in a xenograft model compared to free EGCG [30]. In the in vitro model, the EGCG encapsulated chitosan-coated nanoliposomes (CALIPPO) and EGCG loaded folic acid-poly(ethylene glycol)-FA-PEG nanoparticles induced apoptosis and inhibited MCF-7 breast cancer cells [8,31]. EGCG has been extensively studied as a prospective anti-tumour drug in the treatment of melanoma cancer. The epigallocatechin-3-gallate-loaded fucose-chitosan/polyethylene glycol-chitosan/gelatin nanoparticles induced apoptosis in MKN45-Luc gastric cancer cells and reduced vascular endothelial growth factor protein expression [32].

Other nanoparticles that improve the anticancer effect of EGCG are nanoethosomes and chitin-loaded honokiol. A study by Liao et al. showed that transdermal delivery docetaxel loaded in EGCG-nanoethosomes reduced the tumour volume. The nanoethosomes allow transdermal delivery of flavonoids to melanoma cancer cells [33]. The Tang et al. study showed that EGCG nanoparticles (chitin-laden honokiol) inhibited the proliferation of HepG2 cancer cell cells by inhibiting cells in the G2/M phase and reducing the potential of the mitochondrial membrane [34]. The anticancer effect of epigallocatechin-3-gallate nanoparticles is represented in Table 1.

## 3. Quercetin

Quercetin (QT) is one of the most abundant flavonoids found in vegetables. It causes cell cycle arrest in proliferating lymphoid cells and inhibits growth and formation of several tumour cell lines in vitro [35]. Due to low water solubility and poor absorption of this flavonoid, numerous in vivo and in vitro studies have been conducted on nanoparticles to improve physicochemical properties. In vivo quercetin nanoparticles have a potent anticancer effect on A549 lung cancer cells [9,36], A2780S ovarian cancer cells [24,25], B16F10 melanoma cells [26,27], MCF-7, T1 breast cancer cells [10,36,37,38], CT26 colorectal cancer cells [11], U87 neuroglioma cells [39], C6 glioma cells [40], U14 cervical cancer cells [41], and hepatocellular cancer cells [36,42,43,44,45,46].

The most commonly used nanoparticles with quercetin are PEG, poly(lactic co-glycolic acid) nanoparticles (PLGA) and PLA. It has been reported that PEG nanoparticles prolong the circulation time of quercetin in the bloodstream and increase its solubility and stability [47]. The Tan et al. study showed that the PEG-derivatized phosphatidylethanolamine nanomicelles improved the anticancer activity of quercetin. It has been observed that these nanoparticles were more effective against A549 lung cancer cells than free quercetin [9]. Another study by Xing et al. showed that de-PEGylated nanoparticles based on triphenylphosphine-quercetin (TPP-PEG) were more effective therapeutic agents compared to pure quercetin in A459, MCF-7, and HepG2 cancer cells [36]. The anticancer activity of quercetin loaded in PEG was observed in the study of Dora et al. The results indicated that oral administration of nanosized PEG emulsion containing quercetin had cytotoxicity activity against B16F10 melanoma cells [48]. Similarly, by using quercetin loaded PEG-liposomal nanoparticles, inhibition of angiogenesis of ovarian cancer was achieved [49]. The Zhao et al. study showed that the 1,2-distearoyl-sn-glycero-3-phosphoethanolamine-*N*-methoxy(polyethylene glycol) (DSPE-MPEG) is a good adjuvant nanocarrier for anticancer drug delivery [50]. A few studies on the treatment of hepatocellular cancer with PLGA-loaded quercetin nanoparticles have been shown. In the Ghosh et al. study, the PLGA nanoparticles loaded in quercetin completely protected the mitochondrial membrane of the liver against cancer induced by diethylnitrosamine [42]. After oral administration, PLGA nanoparticles encapsulated with quercetin and tamoxifen (TMX) controlled tumour angiogenesis in MCF-7 breast cancer cells [10]. The Panday et al. study has proven that oral administration rutin-loaded PLGA nanoparticles of quercetin improved hepatic parameters and increased superior inflammatory markers [45]. Quercetin encapsulated in monomethoxy poly(ethylene glycol)-poly(ε-caprolactone) (MPEG-PCL) nanoparticles inhibited ovarian tumour growth by the mitochondrial apoptotic pathway [51,52]. Moreover, the anticancer efficacy of GeluPearl comprising of Precirol ATO 5 lipid (GPSLN) nanoparticles loaded with quercetin against B16F10 melanoma cells was proved [53]. In another study (Mandal et al.), inhibition of development of hepatocellular carcinoma by PLA nanoparticles has been shown [43]. The gold-quercetin in PLA nanoparticles inactivated the caspase/Cyto-c pathway in hepatocellular cells [44]. The da Luz et al. study has shown that poly-lactic acid nanoparticles have an anticancer effect on A549 lung cancer cells [54]. In the Li et al. study, quercetin-loaded soybean phosphatidylcholine-cholesterol (SPC-CHOL) inhibited U14 cervical cancer cells [41]. The anticancer activity of quercetin against breast cancer was observed by using PVP [38]. To improve the effect of quercetin, gold-PLA nanoparticles were also used [39,44,55]. The in vivo studies have confirmed that quercetin nanoparticles have potential in cancer treatment (Table 2).

## 4. Genistein

Genistein is an isoflavonoid found in a number of plants including soybeans, fava beans, and lupins [56]. Some studies have shown beneficial effects of genistein nanoparticles against several cancer lines. Genistein has poor water solubility, rapid metabolism, and low oral bioavailability, which limit the clinical application of this flavonoid [57]. Genistein loaded TPGS-b-PCL (d-α-tocopheryl polyethylene glycol 1000 succinate-poly(ε-caprolactone)) inhibited HeLa cervical tumour cells growth and had a higher level of cytotoxicity compared to genistein-loaded PCL nanoparticles [58], whereas genistein-loaded M-PLGA-TPGS (poly(d,l-lactide-co-glycolide)-d-α-tocopheryl polyethylene glycol 1000 succinate) had a linear apoptotic effect against HepG2 liver cancer cells [12]. Unfortunately, the improvement in the bioavailability of genistein resulting from the use of nanoparticles is associated with high cytotoxicity in normal cells. Further in vitro studies focused on anticancer activity (apoptosis and autophagy of cancer cells) in colon cancer HT29 cells with genistein-loaded PEGylated silica hybrid nanomaterials and lung cancer A549 cells with genistein-miRNA-29b-loaded hybrid nanoparticles-GMLHN, as well as hematopoietic cancer cells with genistein-carboxymethylated chitosan nanoparticles-Fe3O4-CMC [59,60,61]. Despite low water solubility, low bioavailability, and instability of pure genistein, the genistein-loaded nanoparticles described in the presented studies have made it possible to use them in anticancer treatment [62]. Studies on the anticancer activity of genistein nanoparticles are presented in Table 3.

## 5. Silibinin

Silibinin is found in the seeds of milk thistle [63]. Studies show that silibinin has an antineoplastic potential against many cancers by promoting the cell-cycle and inhibiting proliferation [64]. However, due to its hydrophobic structure, it has poor water solubility and permeability across intestinal epithelial cells. To improve the effect of silibinin, nanoparticles, i.e., PEG, polyvinyl alcohol (PVA), and poly-*N*-(2-hydroxypropyl) methacrylamide (pHPMA)-coated wheat germ agglutinin-modified lipid-polymer hybrid nanoparticles have been used [65]. The study of Xu et al. on silibinin nanoparticles described their effect on cancer cells and the blocking of metastasis of breast cancer. These silibinin-loaded lipid nanoparticles (SLNs) containing TPGS and phosphatidylcholine were designed and prepared by a thin-film hydration method [66]. In another study (Gohulkumar et al.), silibinin encapsulated in PVA-Eudragit nanoparticles showed anticancer efficacy in oral carcinoma cells [67]. Another study showed that silibinin encapsulated in PEG nanoparticles had a cytotoxic effect on breast cancer (MCF 10A) in vitro [68]. Free silibinin has low solubility and inadequate dissolution, which cause low oral bioavailability. The Sahibzada et al. study shows two methods for manufacturing nanoparticles of silibinin (APSP—anti-solvent precipitation with a syringe pump and EPN—evaporative precipitation of nanosuspension), which increase its solubility, making this flavonoid a potential oral drug in cancer therapy [69]. Huo et al. showed that the combination therapy of silibinin and paclitaxel (PTX) loaded in dextran-deoxycholic acid (Dex-DOCA) nanoparticles effectively accumulate in tumour sites by passive targeting and inhibit tumour growth through an enhanced intratumoural penetration in mice [70]. Changes in the tumour microenvironment were observed in another study where silibinin and IPI-549 nanoparticles (AEAA-PEG-PCL-aminoethyl anisamide-polyethylene glycol-polycaprolactone) inhibited 4T1 breast cancer cells [71]. It is also worth noting that silibinin demonstrates an anti-metastasis effect. Research on the poly-*N*-(2-hydroxypropyl) methacrylamide (pHPMA)-coated wheat germ agglutinin-modified lipid-polymer hybrid nanoparticles, co-loaded with silibinin and cryptotanshinone (S/C-pW), showed inhibition of tumour growth in 4T1 tumour-bearing mice and presented anti-metastasis activity in the lung [72]. In vitro and in vivo studies concerning the potential anticancer activity of silibinin nanoparticles are presented in Table 4.

## 6. Apigenin

Apigenin is found in several types of vegetables and fruits, especially berries. This flavonoid is involved in regulating signalling pathways in hepatocellular carcinoma and skin cancer. The beneficial effect of free apigenin in cancer treatment is relatively low because of its low lipid and water solubility [73]. Nowadays, to improve the bioavailability of flavonoids, PLGA nanoparticles are mainly used. A study by Das et al. showed an anti-proliferative effect of apigenin loaded in PLGA nanoparticles on A475 skin cancer cells. It is worth emphasizing that these nanoparticles were effective in maintaining photodegradation through ultraviolet light [74]. Other study on apigenin encapsulated in PLGA nanoparticles showed that intravenous administration of this molecules successfully reached HepG2 and Huh-7 cells in vitro as well as the liver of carcinogenic animals and delayed development of hepatocellular carcinoma in rats [75]. Novel nanomaterials have recently been investigated, which contribute to enhanced solubility and bioavailability of apigenin via preparation of solid dispersions of mesoporous silica nanoparticles [76]. The anticancer effects of apigenin nanoparticles are reported in Table 5.

## 7. Naringenin

Naringenin is widely distributed in fruits, especially citrus fruits, bergamot, and tomatoes [77]. Its anticancer properties are associated with anti-inflammatory and antioxidant activities. The clinical effect of cancer treatment with naringenin is limited by its low solubility and minimal bioavailability, related to its hydrophobic ring structure [78].

Current research has shown that naringenin nanoparticles can inhibit carcinogenesis in oral squamous cell carcinoma [79], lung cancer (A549) [80], and colorectal cancer (colon-26) [81]. Due to their anti-proliferative and antioxidant potential, nanoparticles can be potentially useful in oral cancer chemoprevention [79,82]. In the Sulfikkarali et al. study, PVA-EE (polyvinyl alcohol and Eudragit 500) loaded with naringenin had a positive effect on DMBA-induced oral squamous cell carcinoma in hamsters. The oral administration of PVA-EE-naringenin completely prevented the tumour formation as compared to the free naringenin [79]. Chitosan nanoparticles encapsulating naringenin were used in an in vitro lung cancer model (A549), reporting cytotoxic effects on the cancer cells while having a nontoxic effect on normal 3T3 fibroblast cells [80]. The Chaurasia et al. study has described tumour suppression in BALB/c mice bearing colon-26 cells. Both naringenin-encapsulated soluthin-maltodextrin nanoparticles and EE-naringenin improve bioavailability and have a cytotoxic effect against colorectal cancer cells [32,33]. The Fuster et al. study reported that naringenin loaded in silk fibroin nanoparticles had anticancer potential for treatment of cervical cancer HeLa cells [78] (Table 6).

## 8. Luteolin

Luteolin is found in various types of plants such as fruits, vegetables, and medicinal herbs [84]. It induces apoptosis and inhibits cancer cell migration, invasion, and angiogenesis [85]. Due to its hydrophobic structure, it has poor water solubility, poor systemic delivery, and low efficacy. To improve the effect of luteolin, nanoparticles, i.e., PLA-PEG and folic acid-PEG-PCL have been used [86].

One in vitro study concluded that water-soluble polymer-encapsulated nano-luteolin from hydrophobic luteolin (PLA-PEG) inhibited the growth of lung cancer cells (H292 cell line) and squamous cell carcinoma of head and neck (SCCHN) cells (Tu212 cell line) [86], which are among the most frequent cancers worldwide [87,88,89]. Moreover, in vivo study using a tumour xenograft mouse model demonstrated that nano-luteolin has a significant inhibitory effect on the tumour growth of SCCHN in comparison to free luteolin [86]. The study by Wu et al. study showed that luteolin encapsulated in folic acid modifiedpoly(ethylene glycol)-poly(ecaprolactone) (Fa-PEG-PCL) nano-micelles induced glioblastoma multiforme growth of GL261 cells [90]. Additionally, in the safety assessment of nanoparticles used in the tested mice, there were no obvious side effects [90]. Due to the fact that luteolin is hydrophobic [91] and has low biocompatibility [90], it still requires further studies to improve its bioavailability (Table 7).

## 9. Kaempferol

Kaempferol is widely distributed in vegetables (broccoli, spinach), fruits (strawberries, apples), and herbal medicines [92]. The limitation of free kaempferol is inefficient systemic delivery and limited bioavailability [93]. Promising nanoparticles to improve the anticancer efficacy of this flavonoid are chitosan, gold, and PLGA. The potential anticancer effect of this flavonoid is associated with inhibition of phosphatidylinositol-3-kinase (Pl-3) and ribosomal s6 kinase (rsk) and cell cycle arrest in various cancer types [94]. In vitro studies have demonstrated that kaempferol nanoparticles can inhibit carcinogenesis in ovarian cancer cells (A2780/CP70 and OVCAR-3) [93], rat glioma cells (C6) [95] and lung cancer cells (A549) [96]. In the study by Luo et al. both PEO-PPO-PEO as well as PLGA nanoparticles formulations incorporating kaempferol significantly reduced viability of ovarian cancer cells (A2780/CP70 and OVCAR-3), compared with kaempferol alone [93]. The PEO-PPO-PEO nanoparticles were more effective than PLGA nanoparticles, however, PEO-PPO-PEO nanoparticles reduced the viability of ovarian cancer cells (OVCAR-3) and normal ovarian cells (IOSE397), while PLGA nanoparticles had selective toxicity and reduced only the viability of ovarian cancer cells (OVCAR-3) [93]. In another study, kaempferol-loaded mucoadhesive chitosan nanoemulsion (MNE) was used to induce glioma cell in rats [95]. The kaempferol-loaded MNE reduced C6 glioma cell viability a greater degree than free kaempferol. Therefore, kaempferol-loaded mucoadhesive chitosan nanoemulsion could be a promising alternative for brain cancer treatment [95]. The Govindaraju et al. study reported that kaempferol with gold nanoparticles had higher toxicity to A549 lung cancer cells than to normal human cells [96] (Table 8).

## 10. Other Flavonoids (Fisetin and Myricetin)

Fisetin is found in various fruits and vegetables (e.g., apple, strawberry, grape, persimmon, onion) [97]. In the Ghosh et al. study, fisetin-loaded human serum albumin nanoparticles revealed anticancer activity against MCF-7 breast cancer cells in vitro [98]. The Feng et al. study confirmed the effect of fistein nanoparticles on 4T1 breast cancer cells in vivo. Additionally, in vitro fistein-loaded PLA nanoparticles had an anti-tumour effect against colon cancer HCT116 cells [99].

Myricetin is very common in various plants, such as vegetables, fruits, and in teas and wines [100]. In vitro myricetin encapsulated in solid Gelucire-based lipid nanoparticles in the presence of phosphate buffer provided sustained release with no signs of degradation [101]. In the Khorsandi et al. study, solid lipid nanoparticles of myricetin induced growth of lung cancer A549 cells and increased necrosis with no influence on proliferation and apoptosis [102].

The anticancer effects of fisetin and myricetin nanoparticles are presented in Table 9.

Flavonoids, a ubiquitous group of naturally occurring polyphenolic compounds, have recently gained importance as anticancer agents. Unfortunately, due to low solubility, absorption, and rapid metabolism of dietary flavonoids, their anticancer potential is not sufficient. The use of nanotechnology has improved the bioavailability of flavonoids and has increased their anti-tumour activity.

Both in vitro and in vivo studies have shown that flavonoid nanoparticles are promising in cancer treatment in the near future. Most of the studies determining the anticancer properties of flavonoid nanoparticles are preclinical. The potential anticancer activity focuses mainly on MCF-7 breast cancer cells, A549 lung cancer cells, HepG2 liver cancer cells, and melanoma cells.

The anticancer activity of flavonoid nanoparticles is associated with apoptosis and antiproliferation, inhibition of the cell cycle of cancer cells, regulation of the host’s immune system or an anti-inflammatory effect. A particular issue worth investigation is the influence of nanoparticles on the tumour microenvironment, which may be quite important in metastasis.

It is worth noting that flavonoid nanoparticles can also support the anti-tumour effect of drugs used in cancer therapy by enhancing the anti-tumour effect or by reducing the systemic toxicity of drugs.

## Figures and Tables

**Table 1 biomolecules-10-01268-t001:** Anticancer effect of epigallocatechin-3-gallate (EGCG) nanoparticles in in vitro and in vivo studies.

Nanomaterial Type	Cancer Type/Effect	Study Type	Reference
**PLA-PEG**	PROSTATE CARCINOMA over 10-fold dose advantage for exerting its proapoptotic and angiogenesis inhibitory effects on 22Rr1 cells	Animal model (mice)	Siddiqui I. A. et al., 2009 [21]
**Gold**	BLADDER TUMOUR inhibition of tumour growth (MBT-2 cells)	Animal model (mice)	Hsieh D. S. et al., 2011 [23]
	PROSTATE CARCINOMA inhibition of the tumour growth (PC-3 cells)	Animal model (mice)	Shukla R. et al., 2012 [25]
	BLADDER TUMOUR inhibition of tumour growth (MBT-2 cells)	Animal model (mice)	Hsieh D. S. et al., 2012 [24]
	MELANOMA inhibition of tumour growth (B16F10 cells)	Animal model (mice)	Chen C. C. et al., 2014 [26]
**Carbohydrate matrix of gum-arabic and maltodextrin**	PROSTATE CARCINOMA induction of apoptosis, reduction of the cell viability (Du145 cells)	In vitro model	Rocha S. et al.,2011 [27]
**Chitosan**	MELANOMA induction of apoptosis Mel 928 cells	Animal model (mice)	Siddiqui I. A. et al., 2014 [29]
	PROSTATE CARCINOMA inhibitory effect on cancer cells (22Rr1 cells)	Animal model (mice)	Khan N. et al., 2014 [30]
**CSLIPO** **chitosan-coated nanoliposomes**	BREAST CANCER anti-proliferative and proapoptotic effect (MCF-7 cells)	In vitro model	de Pace R. C. C. et al., 2013 [8]
**FA-PEG** **folic acid-poly(ethylene glycol)**	BREAST CANCER inhibition of MCF-7 cell proliferation	In vitro model	Zeng L. et al., 2017 [31]
**FU-PEG** **fucose-chitosan/polyethylene glycol-chitosan/gelatin**	GASTRIC CANCER inhibition of tumour growth (MKN45-Luc cells)	Animal model (mice)	Lin Y. H. et al., 2015 [32]
**EGCG-nanoethosomes**	MELANOMA inhibition of tumour growth (A375 human melanoma cells)	Animal model (mice)	Liao B. et al.,2016 [33]
**Chitin loaded-honokiol**	LIVER CANCER inhibited more cells in the G2/M phase and decreased mitochondrial membrane potential (HepG2 cells)	Animal model (mice)	Tang P. et al., 2018 [34]

**Table 2 biomolecules-10-01268-t002:** Anticancer effect of quercetin (QT) nanoparticles in in vivo studies.

Nanomaterial Type	Cancer Type/Effect	Study Type	Reference
**PEG** **poly(ethylene glycol)**	MELANOMA inhibition of tumour growth (B16F10 melanoma cells)	Animal model (mice)	Dora C. L. et al., 2016 [48]
**PEG-** **phosphatidylethanolamine**	LUNG CANCER anticancer activity in the A549 lung cancer cells	Animal model (mice)	Tan B. J. et al., 2012 [9]
**PEG-liposomal** **polyethylene glycol-liposomal**	OVARIAN CANCER induction of apoptosis and inhibition of angiogenesis	Animal model (mice)	Long Q. et al., 2013 [49]
**TPP-PEG** **Triphenylphosphine** **quercetin nanoparticles poly(ethylene glycol)**	MITOCHONDRIA-TARGETED TUMOUR THERAPY (MCF-7, A459 and HepG2 cells)	Animal model (mice)	Xing L. et al., 2017 [36]
**DSPE-MPEG** **1,2-distearoyl-sn-glycero-3-phosphoethanolamine-*N*-methoxy(polyethylene glycol)**	PROSTATE CANCER apoptosis induction of human androgen-independent PC-3 cells increased drug accumulation at the tumour site and superior anticancer activity	Animal model (mice)	Zhao J. et al., 2016 [50]
**PLGA (poly(lactic co-glycolic acid) nanoparticles**	HEPATOCELLULAR CARCINOMA protection of the mitochondrial membrane of the liver from carcinoma mediated prevention of the cytochrome C expression in the liver	Animal model (rats)	Ghosh A. et al., 2012 [42]
**PLGA-TMX** **(poly(lactic co-glycolic acid) tamoxifen**	BREAST CANCER oral administration efficiently controlled the tumour angiogenesis, normalized levels of the markers (MMP-2 and MMP-9) in MCF-7 cells	Animal model (rats)	Jain A. K. et al., 2013 [10]
**RT-PLGA (poly(lactic co-glycolic acid) nanoparticles of rutin**	HEPATOCELLULAR CARCINOMA reduced incidence of hepatic nodules, necrosis formation, infiltration of inflammatory cells, blood vessel inflammation and cell swelling	Animal model (rats)	Pandey P. et al., 2018 [45]
**QT-loaded PLGA-TPGS (QPTN)** **poly-(dl-lactic-co-glycolic acid)-D-α-tocopherol polyethylene glycol succinate**	LIVER CANCER suppression of the tumour growth HepG2 and HCa-F cells	Animal model (mice)	Guan X. et al., 2016 [46]
**MPEG-PCL** **monomethoxy poly(ethylene glycol)-poly(ε-caprolactone)**	OVARIAN CANCER inhibition of the growth of A2780S ovarian cancer cells through the mitochondrial apoptotic pathway	Animal model (mice)	Gao X. et al., 2012 [51]
	COLORECTAL CANCER improved apoptosis induction and inhibition of cell growth in CT26 cells	Animal model (mice)	Xu G. et al., 2015 [11]
**GPSLN** **GeluPearl comprising of Precirol ATO 5 nanoparticles**	MELANOMA reduced lung colonization and enhanced anti-metastatic activity against B16F10 melanoma cells	Animal model (mice)	Jain A. S. et al., 2013 [53]
**PLA** **poly (dl-lactide-co-glycolide)**	HEPATOCELLULAR CARCINOMA restricted development of hepatocarcinogenesis	Animal model (rats)	Mandal A. K. et al., 2014 [43]
**MPEG-PLA** **methoxy poly(ethylene glycol)-poly(lactide)**	BREAST CANCER inhibition of tumour growth (mammary cancer T1 cells)	Animal model (mice)	Sharma G. et al., 2015 [52]
**Gold-PLA** **gold-quercetin into poly (dl-lactide-co-glycolide) nanoparticles**	CERVICAL CANCER induced apoptosis, autophagy and anti-proliferation via Janus kinase 2 suppression	Animal model (mice)	Luo C. L. et al., 2016 [55]
	NEUROGLIOMA induced autophagy and apoptosis in human neuroglioma U87 cells through activation LC3/ERK/Caspase-3 and suppression of AKT/mTOR signaling pathway	Animal model (mice)	Lou M. et al., 2016 [39]
	HEPATOCELLULAR CARCINOMA inactivation of caspase/Cyto-c pathway, suppression of AP-2β/telomerase reverse transcriptase hTERT, inhibition of NF-κB/cyclooxygenase 2 COX-2 and Akt/ERK1/2 signaling pathways	Animal model (mice)	Ren K. W. et al., 2017 [44]
**Freeze-dried polymeric micelles**	GLIOMA cytotoxic effect on C6 glioma cells	Animal model (mice)	Wang G. et al., 2016 [40]
**SPC-CHOL** **soybean phosphatidylcholine-cholesterol**	CERVICAL CANCER in vitro anti-tumour efficacy to Hela cells in vivo inhibition effect on U14 cells	Animal model (mice)	Li J. et al., 2017 [41]
**PVP** **poly (vinyl pyrrolidone)**	BREAST CANCER antioxidative activity and efficient photothermal killing effect to cancer 4T1 cells	Animal model (mice)	Tang S. H. et al., 2019 [38]

**Table 3 biomolecules-10-01268-t003:** Anticancer effect of genistein nanoparticles in in vitro and in vivo studies.

Nanomaterial Type	Cancer Type/Effect	Study Type	Reference
**TPGS-b-PCL**	CERVICAL CANCER inhibition of tumour growth (HeLa cells)	Animal model (mice)	Zhang H. et al., 2015 [58]
**M-PLGA-TPGS**	LIVER CANCER apoptotic effect against HepG2 cells	Animal model (mice)	Wu B. et al., 2016 [12]
**PEGylated silica hybrid nanomaterials**	COLON CANCER HT29 cells modulation of endogenous antioxidant enzymes and H_2_O_2_ production, which simultaneously activated apoptosis and autophagy	In vitro model	Pool, H et al., 2018 [59]
**GMLHN** **(Genistein-miRNA-29b-loaded hybrid nanoparticles)**	LUNG CANCER Phosphorylated protein kinase, strain AK, Thymoma (Phosphorylated protein kinase B) (pAKT), Phosphorylated phosphoinositide 3-kinase (p-PI3K), DNA (cytosine-5-)-methyltransferase 3 beta (DNMT3B) and Myeloid Cell Leukemia Sequence 1 (MCL 1) efficiently downregulated, anti-proliferative effect non-small cell lung cancer A549 cells	In vitro model	Sacko K. et al., 2019 [61]
**Fe_3_O_4_-CMC** **(carboxymethylated chitosan)**	HEMATOPOIETIC CANCER significant growth inhibition of hematopoietic cancer cells	In vitro model	Ghasemi Goorbandi R. et al., 2020 [62]

**Table 4 biomolecules-10-01268-t004:** Anticancer effect of silibinin nanoparticles in in vitro and in vivo studies.

Nanomaterial Type	Cancer Type/Effect	Study Type	Reference
**PVA-Eudragit**	ORAL CARCINOMA inhibition of apoptosis KB cells	In vitro model	Gohulkumar M. et al., 2014 [67]
**PEG**	BREAST CANCER cytotoxic effect on MCF 10A	In vitro model	Sajjadiyan S. Z. et al., 2016 [68]
**AEAA-PEG-PCL**	BREAST CANCER 4T1 cells angiogenesis suppression	Animal model (mice)	Jiang M. et al., 2020 [71]
**PTX-SB-Dex-DOCA**	LUNG CANCER inhibition of tumour growth A549 cells	Animal model (mice)	Huo M. et al. 2020 [70]
**S/C-pW nanoparrticles**	BREAST CANCER anti-metastasis activity	Animal model (mice)	Liu Y. et al., 2020 [72]

**Table 5 biomolecules-10-01268-t005:** Anticancer effect of apigenin nanoparticles in in vivo studies.

Nanomaterial Type	Cancer Type/Effect	Study Type	Reference
**PLGA**	SKIN CANCER A375 cells reduction in markers of proliferative activity increased ROS production and mitochondrial-induced apoptosis	Animal model (mice)	Das S. et al., 2013 [74]
	HEPATOCELLULAR CARCINOMA inhibition effect on cancer cells HepG2 and Huh-7 cells	Animal model (rats)	Bhattacharya S. et al., 2018 [75]
**MSN** **mesoporous silica nanoparticles**	improved solubility, dissolution, and bioavailability after oral application	Animal model (rats)	Huang Y., et al., 2019 [76]

**Table 6 biomolecules-10-01268-t006:** Anticancer effect of naringenin nanoparticles in in vivo studies.

Nanomaterial Type	Cancer Type/Effect	Study Type	Reference
**PVA-eudragit** **poly vinyl alcohol-Eudragit 500**	ORAL SQUAMOUS CELL CARCINOMA anti-tumour effect	Animal model (hamster)	Krishnakumar N. et al., 2013 [79]
**Eudragit**	ORAL SQUAMOUS CELL CARCINOMA prevention of the tumour formation, reduction of the degree of histological lesions, anti-lipid peroxidative effect, anti-proliferative effect, antioxidant potential	Animal model (hamster)	Sulfikkarali N. et al., 2013 [82]
**Chitosan**	LUNG CANCER antioxidant and anticancer activities (A549 cells)	In vitro model	Kumar S. P. et al., 2015 [80]
**Soluthin-maltodextrin**	COLORECTAL CANCER enhanced oral bioavailability, tumour suppression in BALB/c mice-bearing colon-26 cells	Animal model (rats)	Chaurasia S. et al., 2017 [81]
**NRG-EE100-NPs** **(Eudragit 100)**	COLORECTAL CANCER enhanced oral bioavailability, tumour suppression in BALB/c mice-bearing colon-26 cells	Animal model (rats)	Chaurasia S. et al., 2018 [83]
**Silk fibroin nanoparticles**	CERVICAL CANCER anticancer potential HeLa cells	In vitro model	Fuster M. G. et al., 2020 [78]

**Table 7 biomolecules-10-01268-t007:** Anticancer effect of luteolin nanoparticles in in vivo studies.

Nanomaterial Type	Cancer Type/Effect	Study Type	Reference
**PLA-PEG**	LUNG CANCER and HEAD AND NECK CANCER inhibition of tumour growth H292, Tu212 cells	Animal model (mice)	Majumdar D. et al., 2014 [86]
**Fa-PEG-PCL**	GLIOBLASTOMA multiforme induced cell growth inhibition and apoptosis of GL261 cells	Animal model (mice)	Wu C. et al., 2019 [90]

**Table 8 biomolecules-10-01268-t008:** Anticancer effect of kaempferol nanoparticles in in vitro studies.

Nanomaterial Type	Cancer Type/Effect	Study Type	Reference
**PEO-PPO-PEO, PLGA**	OVARIAN CANCER strong and selective inhibition of cancer cell viability A2780/CP70 and OVCAR-3 ovarian cancer cell lines	In vitro model	Luo H. et al., 2012 [93]
**Chitosan**	BRAIN CANCER induction of apoptosis C6 rat glioma cell line	In vitro model	Colombo M. et al., 2018 [95]
**Gold**	LUNG CANCER cytotoxic effect on A549 lung cancer cells	In vitro model	Govindaraju S. et al., 2019 [96]

**Table 9 biomolecules-10-01268-t009:** Anticancer effect of fisetin and myricetin nanoparticles in vitro and in vivo studies.

Flavonoid	Nanomaterial Type	Cancer Type/Effect	Study Type	Reference
**Fisetin**	**HAS (human serum albumin)**	BREAST CANCER cytotoxic effect on MCF-7 cells	In vitro model	Ghosh P. et al., 2016 [98]
	**PLA**	BREAST CANCER and COLON CANCER cytotoxicity assay against HCT116 colon cancer cells in vitro and anti-tumour test in a xenograft 4T1 breast cancer model in vivo demonstrated the anti-tumour effect	In vitro and animal model (rats)	Feng C. et al., 2019 [99]
**Myricetin**	**SLNs solid lipid nanoparticles**	LUNG CANCER induction of cell growth necrosis (A549 cells)	In vitro model	Khorsandi L. et al., 2020 [102]

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
