# Peer review of "Flavonoid Nanoparticles: A Promising Approach for Cancer Therapy"

_biomolecules, 2020, doi:10.3390/biom10091268_

Round 1

Reviewer 1 Report

Dobrzynska and her coworkers collected the online available sources about some flavonoid-loaded nanoparticles with anti-cancer properties. In the title they mention antioxidant effect, in the manuscript, however, this feature remains mainly uncovered. Although they made a huge work by collecting and classifying the cited references, I can see only a limited added value: this manuscript seems to be a simple enumeration, and in certain parts, it gives the impression of a list of abbreviations. In some cases, the description cannot be understood without checking the cited reference (especially for ref. 32., 44).

I could suggest to the authors rewriting their text, choosing a novel conception, and explaining the importance of the cited studies. E.g. classification according to the type of the nanoparticles could be intriguing: they found a method in which the native flavonoid was used to form nanoparticles, which showed improved solubility and other advantageous properties; in the other cases different additives were applied, such as carbohydrates, PEG, gold NP, etc. These could be compared and analyzed in light of the findings for the different flavonoids. Another important class is when the flavonoids are covalently conjugated with other molecules (e.g. ref. 32.) leading to self-assembly. Here I would mention, that curcumin is not a flavonoid, albeit it may have similar physiological effects, thus it can be omitted from this compilation.

The authors followed carefully the journal’s instructions for citing the references and formatting, however, in a review article the sections ’Materials and Methods’ and ’Results’ are not applicable. The English of the text was rather poor.

In summary, I do not suggest accepting this manuscript for publication in Biomolecules, the authors should reconstruct it thoroughly.

Author Response

Reviewer 1:

Firstly, we would like to express our deepest thanks to the Reviewer for devoting time to reviewing our manuscript, the corrections and suggestions. We have carried out a major revision of the manuscript and we believe the paper has been significantly improved.

The reviewer's comment: The Authors collected the online available sources about some flavonoid-loaded nanoparticles with anti-cancer properties. In the title they mention antioxidant effect, in the manuscript, however, this feature remains mainly uncovered. Although they made a huge work by collecting and classifying the cited references, I can see only a limited added value: this manuscript seems to be a simple enumeration, and in certain parts, it gives the impression of a list of abbreviations. In some cases, the description cannot be understood without checking the cited reference (especially for ref. 32., 44).

The authors’ answer: According to the Reviewer's suggestion, the changes have been made in the manuscript and we enriched it by necessary information.

The reviewer's comment: I could suggest to the authors rewriting their text, choosing a novel conception, and explaining the importance of the cited studies. E.g. classification according to the type of the nanoparticles could be intriguing: they found a method in which the native flavonoid was used to form nanoparticles, which showed improved solubility and other advantageous properties; in the other cases different additives were applied, such as carbohydrates, PEG, gold NP, etc. These could be compared and analyzed in light of the findings for the different flavonoids. Another important class is when the flavonoids are covalently conjugated with other molecules (e.g. ref. 32.) leading to self-assembly. Here I would mention, that curcumin is not a flavonoid, albeit it may have similar physiological effects, thus it can be omitted from this compilation.

The authors’ answer: We are thankful for this important comment. We have re-arrange the text of the manuscript in accordance to the Reviewer's suggestion. We added relevant information about nanoparticles and classification, and we removed the description of the curcumin.

The reviewer's comment:. The authors followed carefully the journal’s instructions for citing the references and formatting, however, in a review article the sections ’Materials and Methods’ and ’Results’ are not applicable. The English of the text was rather poor.

The authors’ answer: We are thankful for this comment. According to the Reviewer's suggestion, the changes have been made in the manuscript and highlighted by colour (Lines: 51-53, 58-61, 62-67, 73-75, 77-79, 88-89, 90-91, 110-112, 115-119,121-124,128-130,136-137,139-142, 154-156, 177-182, 209-211, 228-230, 255-256, 275-277). The manuscript has been checked by a Native-speaker.

Reviewer 2 Report

This manuscript is interesting but two major conerns should be considered.

1. Authors have to indicate why flavonoids have weak points in terms of absorption or metabolism or tissue distribution etc.

Also, each flavonoid has different drawbacks and then how nanoparticles overcome those drawbacks has to be described.

Instead of simply mentioning in vivvo bioactivities are improved, how those improvement could be achieved has to be mentioned.

2. Each nano materials would have some properties. At least, why these materials are selected for individual compounds has to be mentioned.

Author Response

Reviewer 2

Firstly, we would like to express our deepest thanks to the Reviewer for devoting time to reviewing our manuscript, the corrections and suggestions. We have carried out a major revision of the manuscript and we believe the paper has been significantly improved.

The reviewer's comment: Authors have to indicate why flavonoids have weak points in terms of absorption or metabolism or tissue distribution etc. Also, each flavonoid has different drawbacks and then how nanoparticles overcome those drawbacks has to be described. Instead of simply mentioning in vivo bioactivities are improved, how those improvement could be achieved has to be mentioned.

The authors’ answer: We are thankful for this important comment. According to the Reviewer's suggestion, the changes have been made in the manuscript. We have completed the necessary information about absorption, metabolism and distribution of flavonoids and how nanoparticles improve bioactivity of flavonoids.

The reviewer's comment: Each nano materials would have some properties. At least, why these materials are selected for individual compounds has to be mentioned.

The authors’ answer: According to the Reviewer's suggestion, the changes have been made in the manuscript and highlighted by colour (Lines: 51-53, 58-61, 62-67, 73-75, 77-79, 88-89, 90-91, 110-112, 115-119,121-124,128-130,136-137,139-142, 154-156, 177-182, 209-211, 228-230, 255-256, 275-277).

Reviewer 3 Report

In this review the authors aimed to evaluate studies on the anticancer activity of flavonoid nanoparticles. The manuscript describes the anticancer activity of the most important flavonoids delivered by nanoparticles and for each flavonoid the type of nanoparticles and the targeting tumor was reported.

 the cited review :  Semin Cancer Biol 2019, S1044579X19301828. doi:10.1016/j.semcancer.2019.07.023 should be reported at the end of the sentence of line  36.

Author Response

Reviewer 3

Firstly, we would like to express our deepest thanks to the Reviewer for devoting time to reviewing our manuscript, the corrections and suggestions. We have carried out a major revision of the manuscript and we believe the paper has been significantly improved.

The reviewer's comment: In this review the authors aimed to evaluate studies on the anticancer activity of flavonoid nanoparticles. The manuscript describes the anticancer activity of the most important flavonoids delivered by nanoparticles and for each flavonoid the type of nanoparticles and the targeting tumor was reported. The cited review: Semin Cancer Biol 2019, S1044579X19301828. doi:10.1016/j.semcancer.2019.07.023 should be reported at the end of the sentence of line  36.

The authors’ answer: We appreciate the positive feedback from the Reviewer. According to the Reviewer's suggestion, the changes have been made in the manuscript (Line 36).

Round 2

Reviewer 1 Report

The authors have improved their manuscript to a certain extent. In my opinion, however, this review still does not reach the high standard of this journal.

I have qualms about the followings:

  1. The title is still misleading, the text does not deal with the antioxidant effect of flavonoid nanoparticles.
  2. The text is still poor in English, e.g.:

lines 79-81: Further research was performed on prostate cancer cells where EGCG encapsulated chitosan significant inhibition of tumour growth was observed compared to free EGCG [30].

lines 117-118: The anticancer activity of PEG loaded in quercetin was observed in Dora et. al. study. They results has proven,…

Actually EGCG and quercetin were loaded in chitosan and PEG NPs, respectively.

lines 166-167: The above studies using various nanoparticles have made it possible to use them in anti-cancer treatment despite being limited by its low water solubility, low bioavailability, and instability compared to pure genistein.

There are many similar poorly worded sentences throughout the text.

  1. The text is not in accordance with the cited references, e.g.:

line 121: The Zhao et. al. studies showed that the 1,2-distearoyl-sn-glycero-3-phosphoethanolamine-N-methoxy(polyethylene glycol) (DSPE-MPEG) induced PC-3 prostate cancer cells.

This is not true, the cited literature states that this material is an FDA-approved pharmaceutical adjuvant nanocarrier for anticancer drug delivery.

line 181: The first studies on silibinin nanoparticles described its effect on cancer cells.

No reference is given.

lines 189-198: Currently, we have few Huo et. al. in vivo studies assessing the effect of the silibinin nanoparticles on cancer cells. It has been shown that the combination therapy of silibinin and paclitaxel (PTX) nanoparticles effectively inhibit lung tumour and may modulate of tumour
microenvironment [69].

In the cited article, a dextran-based nanoparticle was used as a carrier, paclitaxel was the active drug delivered together with silibinin.

Table 4 and lines 186-189: The Sahibzada et. al. study have shown two nanoparticles of silibinin (APSP – antisolvent precipitation with a syringe pump and EPN – evaporative precipitation of nanosuspension) increase its solubility which makes this flavonoid a potential oral drug in cancer therapy [68].

From this composition, especially in the Table, it is hard to understand that APSP and EPN are not nanoparticles or additives, but methods for the preparation of nanosized aggregates of free silibinin.

In some cases, abbreviations are not explained.

  1. Editing of the text was not conscientious, e.g.:

line 128: PLGA-TMX loaded quercetin nanoparticles after oral administration [10].

This is an incomplete sentence.

line 136-137: It is worth mentioning that poly-lactic acid nanoparticles are able to maintain therapeutic drug levels for sustained periods of time [54].

This sentence appears quite accidentally at the end of a paragraph about the quercetin-loaded PLA NPs. Similarly, lines 138-139 contain a general statement about bioavailability without any connection to the context.

In summary, I do not suggest accepting this manuscript for publication in Biomolecules.

Author Response

Firstly, we would like to express our deepest thanks to the Reviewer for devoting time to reviewing our manuscript, the corrections and suggestions. We have carried out a major revision of the manuscript and we believe the paper has been significantly improved.

The reviewer's comment: 1. The title is still misleading, the text does not deal with the antioxidant effect of flavonoid nanoparticles.

The authors’ answer: According to the Reviewer's suggestion the title has been changed.

The reviewer's comment: 2. The text is still poor in English, e.g.: lines 79-81: Further research was performed on prostate cancer cells where EGCG encapsulated chitosan significant inhibition of tumour growth was observed compared to free EGCG [30]. lines 117-118: The anticancer activity of PEG loaded in quercetin was observed in Dora et. al. study. They results has proven,… Actually EGCG and quercetin were loaded in chitosan and PEG NPs, respectively. lines 166-167: The above studies using various nanoparticles have made it possible to use them in anti-cancer treatment despite being limited by its low water solubility, low bioavailability, and instability compared to pure genistein. There are many similar poorly worded sentences throughout the text.

The authors’ answer: We are thankful for this important comment. The manuscript has been checked by a special translation office (LIDEX Sp. z o.o., https://www.lidex.com.pl/). We attached the certificate confirming the translation of our manuscript.

The reviewer's comment: 3. The text is not in accordance with the cited references, e.g.: line 121: The Zhao et. al. studies showed that the 1,2-distearoyl-sn-glycero-3-phosphoethanolamine-N-methoxy(polyethylene glycol) (DSPE-MPEG) induced PC-3 prostate cancer cells. This is not true, the cited literature states that this material is an FDA-approved pharmaceutical adjuvant nanocarrier for anticancer drug delivery. line 181: The first studies on silibinin nanoparticles described its effect on cancer cells. No reference is given. lines 189-198: Currently, we have few Huo et. al. in vivo studies assessing the effect of the silibinin nanoparticles on cancer cells. It has been shown that the combination therapy of silibinin and paclitaxel (PTX) nanoparticles effectively inhibit lung tumour and may modulate of tumour microenvironment [69]. In the cited article, a dextran-based nanoparticle was used as a carrier, paclitaxel was the active drug delivered together with silibinin. Table 4 and lines 186-189: The Sahibzada et. al. study have shown two nanoparticles of silibinin (APSP – antisolvent precipitation with a syringe pump and EPN – evaporative precipitation of nanosuspension) increase its solubility which makes this flavonoid a potential oral drug in cancer therapy [68]. From this composition, especially in the Table, it is hard to understand that APSP and EPN are not nanoparticles or additives, but methods for the preparation of nanosized aggregates of free silibinin. In some cases, abbreviations are not explained.

The authors’ answer: We would like to thank the Reviewer for careful review of our manuscript and for providing us with some suggestions to improve it. According to the Reviewer's suggestion, the changes have been made in the manuscript and highlighted by colour.

The reviewer's comment: 4. Editing of the text was not conscientious, e.g.: line 128: PLGA-TMX loaded quercetin nanoparticles after oral administration [10]. This is an incomplete sentence. line 136-137: It is worth mentioning that poly-lactic acid nanoparticles are able to maintain therapeutic drug levels for sustained periods of time [54]. This sentence appears quite accidentally at the end of a paragraph about the quercetin-loaded PLA NPs. Similarly, lines 138-139 contain a general statement about bioavailability without any connection to the context.

The authors’ answer: We are thankful for this comment. According to the Reviewer's suggestion, the changes have been made in the manuscript and highlighted by colour.

Reviewer 2 Report

Now it is acceptable but typographical errors should be corrected.

Author Response

Firstly, we would like to express our deepest thanks to the Reviewer for devoting time to reviewing our manuscript, the corrections and suggestions. We have carried out a major revision of the manuscript and we believe the paper has been significantly improved.

The Reviewer's comment: Now it is acceptable but typographical errors should be corrected.

The Authors’ answer: We appreciate the positive feedback from the Reviewer. According to the Reviewer's suggestion, the changes have been made in the manuscript
